# Self-reported high-risk behavior among first-time and repeat replacement blood donors; a four-year retrospective study of patterns

Felix Osei-Boakye[1,2]*, Charles Nkansah[2,3], Samuel Kwasi Appiah[2,3], Gabriel Abbam[3], Charles Angnataa Derigubah[2,4], Boniface Nwofoke Ukwah[2], Victor Udoh Usanga[2], Emmanuel Ike Ugwuja[5], Ejike Felix Chukwurah[2]

1 Department of Medical Laboratory Science, Faculty of Applied Science and Technology, Sunyani Technical University, Sunyani, Ghana, 2 Department of Medical Laboratory Science, Faculty of Health Science and Technology, Ebonyi State University, Abakaliki, Nigeria, 3 Department of Haematology, School of Allied Health Sciences, University for Development Studies, Tamale, Ghana, 4 Department of Medical Laboratory Technology, School of Applied Science and Arts, Bolgatanga Technical University, Bolgatanga, Ghana, 5 Department of Biotechnology, Faculty of Science, Ebonyi State University, Abakaliki, Nigeria

* foseiboakye1@gmail.com, felix.osei-boakye@stu.edu.gh

## Abstract

### Background

There is no replacement for blood, and patients requiring transfusion depend on human donors, most of whom are family donors. Family donors may deny engagement in high-risk activities, which threaten the safety of donated blood. This study determined frequency of self-reported high-risk behaviors among replacement donors.

### Methods

This retrospective study recruited 1317 donor records from 2017–2020, at Mankranso Hospital, Ghana. Data from archived donor questionnaires were extracted and analyzed with SPSS and GraphPad. Frequencies, associations, and quartiles were presented.

### Results

The donors were predominantly males (84.4%), 17–26 years old (43.7%), informal workers (71.8%), rural inhabitants (56.5%), first-time (65.0%), and screened in the rainy season (56.3%). Donation frequency was significantly associated with age, sex, occupation, and residence. Repeat donors were significantly older ($p \leq 0.001$). More males than females were deferred ($p = 0.008$), drug addicts ($p = 0.001$), had body modifications ($p = 0.025$), multiple sexual partners ($p = 0.045$), and STIs ($p \leq 0.001$), whereas, more females were recently treated ($p = 0.044$). Weight loss ($p = 0.005$) and pregnancy ($p = 0.026$) were frequent among 17–26-year group, whereas, tuberculosis was frequent among 37–60-year group ($p = 0.009$). More first-time donors were unwell ($p = 0.005$), deferred ($p \leq 0.001$), pregnant ($p = 0.002$), drug addicts, had impending rigorous activity ($p = 0.037$), body modifications ($p = 0.001$), multiple sexual partners ($p = 0.030$), and STIs ($p = 0.008$). STIs were frequent in the dry season

**Data Availability Statement:** All relevant data are within the manuscript and its Supporting information files.

**Funding:** The author(s) received no specific funding for this work.

**Competing interests:** The authors have declared that no competing interests exist.

($p = 0.010$). First-time donors had reduced hemoglobin ($p = 0.0032$), weight ($p = 0.0003$), and diastolic pressure ($p = 0.0241$).

## Conclusion

Donation frequency was associated with age, sex, occupation, and residence, with first-time donors younger than repeat donors. Deferral from donation, drug addiction, body modification, multiple sexual partners, and STIs were frequent among males, whereas, more females received treatment. Tuberculosis was frequently reported among older adults, whereas, weight loss and pregnancy were frequent among younger individuals. More first-time donors reported being unwell, deferred, drug addiction, body modifications, multiple sexual partners, STIs, and pregnant. Hemoglobin, weight, and diastolic BP were reduced among first-time donors.

## Introduction

Blood transfusion is a therapeutic and lifesaving procedure useful in conditions that require whole blood, or blood-derived products transfusion, particularly for people with extremely reduced blood volumes and consequent hypoxia. Overall, the aim of blood transfusion services is to provide safe blood and to restore the lives of patients. Currently, there is no true replacement for blood, and patients requiring blood transfusion therapy depend on altruistic human donors for survival [1]. However, this process is not without challenges, as the propensity to spread infections through transfusing human blood persist [1, 2]. In all safety of the patient as well as donor protection are of immense clinical importance in transfusion medicine [3].

Worldwide, about 118.5 million units of blood are donated annually [4], whereas, only 2 million blood donations are made in sub-Saharan Africa [4, 5]. This resulting deficit in blood supply, and the increased demand for hemotransfusions in sub-Saharan Africa is due to increased morbidity in expectant mothers, malaria [1], nutritional insufficiencies [6], deranged hemostasis, hemoglobinopathies, and patients with trauma. In order to meet the increased demands and mitigate the shortages in blood supply, many blood centers in Ghana rely mainly on family replacement blood donors. However, replacement donors are a potential source for the spread of blood-borne infections [7]. Consequently, the National Blood Service Ghana (NBSG), has instituted a more stringent donor recruitment criteria to improve on the existing hemovigilance practices.

The algorithm for screening blood donors recommended by the NBSG, includes a preliminary physical assessment of the donors, followed by the administration of a self-deferral health questionnaire, intended to either temporarily defer or permanently disqualify high-risk and unsuitable donors. Subsequently, serological screening for infectious biomarkers, including, hepatitis B surface antigen (HBsAg), and antibodies to hepatitis C virus, *Treponema pallidum*, and human immunodeficiency virus (HIV) is performed. The aim of these multifaceted selection criteria is to protect individuals who are unfit to donate blood, exclude infected blood, and minimize infections that may be missed by the serological assays during the window period. Despite the safety measures recommended by the NBSG, the screening of blood donors in many blood centers in Ghana is skewed towards the identification of serological biomarkers, while clinical history, anthropometric variables, and lifestyle indicators which may provide early warning signs for blood donor recruiters remain underutilized. Also, the screening for

infections may be improperly performed due to scarce resources [1, 6]. Furthermore, family donors may conceal useful information, including denying their engagement in high-risk activities [8], which are contraindications for blood donation. These practices, in turn, threaten the safety of the donors, donated blood units, and the recipients of blood transfusions. Furthermore, there is paucity of literature on the frequency and patterns of high-risk donor behaviors in Ghana. Therefore, it is imperative to establish evidence to justify the need to educate the public on blood donation and blood donor practices to safeguard the blood supply chain.

Therefore, this study determined the frequency and patterns of self-reported high-risk behaviors among first-time and repeat replacement blood donors. Documenting this data will provide blood centers in Ghana with a baseline information, to: help identify high-risk donor groups, determine the effect of sociodemographic factors on high-risk donor practices, and to unravel the relationship between donation frequency, blood hemoglobin level and some anthropometric variables. Also, this study would help to understand the health educational needs of prospective blood donors in the study setting.

## Participants and methods

### Study design and setting

The present study was a four-year (2017–2020) retrospective cross-sectional study conducted at the Mankranso Government Hospital. In this study, we retrospectively reviewed and collected archived data spanning 2017 and 2020. And, this was done as a onetime sampling conducted between 14th March and 4th April 2024. The hospital offers healthcare to both outpatients and inpatients who seek its services, and has an inpatient capacity of about 88 beds [9]. The hospital has separate admission wards for males, females, children, and another for obstetric- and neonatal patients. Furthermore, the hospital offers services like clinical laboratory, x-ray, ultrasound, antenatal care, and therapy for patients requiring antiretroviral, hypertension, and antidiabetic medications. Also, it operates a surgical theatre where caesarean section and other minor surgeries are performed. The hospital is located at Mankranso in the Ahafo-Ano Southwest district of Ashanti Region, Ghana. It is the major public hospital in the district, and is regulated by the Ghana Health Service. Mankranso is a rural community and is the administrative capital of the Ahafo-Ano Southwest constituency.

### Study population

In this study, all prospective family replacement donors who visited the clinical laboratory unit of Mankranso Government Hospital to undergo screening for the purpose of blood donation were considered. Overall, information was extracted from 1317 clinical records of replacement blood donors who visited the laboratory during a four-year period (1st January 2017 to 31st December 2020). The participants were between the ages of 17 and 60 years.

**Sample size determination.** The computation of sample size in this study, was done with the Epi Info v7.2 epidemiological software (Centers for Disease Control and Prevention [CDC], United States). A previous study from Ghana by Seidu *et al.*, [10] observed 6.0% prevalence of self-reported sexually transmitted infections (STIs), which was used as the percentage for unexposed group in this study. With a two-sided confidence interval of 95%, statistical power of 80%, 1:2 ratio of unexposed-to-exposed, and assumed percentage of 12% was used for the exposed group. This produced the largest sample size of 905 participants (unexposed: 302, and exposed: 603) using the Fleiss w/CC value, and this was increased to 1317 to include all eligible replacement donor records to enhance the representation of the study population.

## Data collection

The National Blood Service Ghana (NBSG) blood donor clinical records (form: NBTS/DN 20 v2) were retrieved from the laboratory's archive, from 14th to 17th March 2024. Pre-donation screening data on the blood donors were then extracted on to a spreadsheet. Blood donors' data extracted from the questionnaire included hemoglobin level; anthropometric data (including weight, systolic blood pressure [SBP], diastolic blood pressure [DBP]); and sociodemographic data (like sex, age, occupation, residence, and season). However, personal identifiers like names were not retrieved from the registers in order to ensure anonymity of the donors. Furthermore, information on blood donation history, and high-risk behaviors were collected. Some of the risky behaviors captured included history of unprotected sex with multiple sexual partners, history of sex with a homosexual, history of needle-stick injury, history of receiving injections outside a hospital, body modifications like tattoos and tribal cutting, history of self-injected drug use, and history of jaundice. Additionally, information on history of chronic and non-chronic diseases like sickle cell disease (SCD), epilepsy, tuberculosis, hepatitis, gonorrhea, human immunodeficiency virus (HIV), and or syphilis were collected.

## Ethical consideration and informed consent

The protocol for the study was approved by the Committee on Human Research, Publication and Ethics (CHRPE) of the School of Medicine and Dentistry, Kwame Nkrumah University of Science and Technology (Reference: CHRPE/AP/177/24). Also, permission was obtained from the authorities of the Mankranso Government Hospital before conducting the study. However, due to the retrospective design of the study, consent from the subjects was not required.

## Statistical data analysis

The data were entered directly into IBM SPSS Statistics for Windows, version 27.0.1.0. (Armonk, NY: IBM Corp.). Continuous data were assessed for skewness using the Kolmogorov-Smirnov test, and subsequently presented as median and interquartile ranges. Age was further transformed into three groups, and presented as frequencies and proportions, along with other categorical data like sex, occupation, frequency of donation, residence, and season. The association of high-risk behavior with sex, donation frequency, and season were determined using the Fisher's Exact test, whereas, the association between high-risk behavior and age was performed with the Chi-Square test. Hemoglobin level, body weight, SBP, and DBP were stratified by donation frequency and further presented as boxplots and whiskers showing the median, lower-, and upper quartiles. The boxplots were visualized using GraphPad Prism 8.4.3 (GraphPad Software, San Diego, California USA). All analysis were considered significant at $p \leq 0.05$.

## Results

### Sociodemographic characteristics of the prospective blood donors

Table 1 shows the sociodemographic characteristics of the prospective family-replacement blood donors stratified by frequency of donation. The median age of the donors was 28.0 (23.0–35.0) years, with the repeat donors significantly older than first-time donors (30.0 [25.0–36.0] vs 27.0 [22.0–34.0], $p \leq 0.001$). Of the 1317 blood donor data reviewed, the majority 1111 (84.4%) were males, 575 (43.7%) were within 1–26-year age range, 945 (71.8%) worked in the informal sector, 744 (56.5%) were rural inhabitants, 741 (56.3%) were screened in the rainy season, and 856 (65.0%) were first-time donors.

**Table 1. Sociodemographic characteristics of prospective replacement donors, stratified by donation frequency, at the Mankranso Government Hospital (2017–2020).**

| Variable | Total screened | Donation frequency | | p |
|---|---|---|---|---|
| | | First-time | Repeat | |
| Age (years) | 28.0 (23.0–35.0) | 27.0 (22.0–34.0) | 30.0 (25.0–36.0) | ≤0.001 |
| **Age group** | | | | ≤0.001 |
| 17–26 | 575 (43.7) | 415 (48.5) | 160 (34.7) | |
| 27–36 | 476 (36.1) | 288 (33.6) | 188 (40.8) | |
| 37–60 | 266 (20.2) | 153 (17.9) | 113 (24.5) | |
| **Sex** | | | | 0.001 |
| Male | 1111 (84.4) | 702 (82.0) | 409 (88.7) | |
| Female | 206 (15.6) | 154 (18.0) | 52 (11.3) | |
| **Occupation** | | | | ≤0.001 |
| Informal | 945 (71.8) | 645 (75.4) | 300 (65.1) | |
| Formal | 161 (12.2) | 69 (8.1) | 92 (20.0) | |
| Student | 148 (11.2) | 102 (11.9) | 46 (10.0) | |
| Unemployed | 63 (4.8) | 40 (4.7) | 23 (5.0) | |
| **Residence** | | | | ≤0.001 |
| Rural | 744 (56.5) | 524 (61.2) | 220 (47.7) | |
| Urban | 573 (43.5) | 332 (38.8) | 241 (52.3) | |
| **Season** | | | | 0.684 |
| Rainy season | 741 (56.3) | 478 (55.8) | 263 (57.0) | |
| Dry season | 576 (43.7) | 378 (44.2) | 198 (43.0) | |
| Total | 1317 (100.0) | 856 (65.0) | 461 (35.0) | |

Age (in years) is presented as median (25th–75th percentiles); Age group, Sex, Occupation, and Residence are presented as frequencies with corresponding proportions in parentheses; Pearson Chi-Square and Fisher's Exact tests were used to compare proportions between first-time and repeat family replacement donors; Mann-Whitney U test was used to compare distribution of Age in years between first-time and repeat family replacement donors; $p \leq 0.05$ was considered significant.

Of the 856 first-time donors, the majority (702 [82.0%]) were males, 415 (48.5%) were 17–26 years of age, 645 (75.4%) worked in the informal sector, and 524 (61.2%) were rural inhabitants. Of the 461 prospective repeat blood donors, 409 (88.7%) were males, 188 (40.8%) were 27–36 years of age, 300 (65.1%) worked in the informal sector, and 241 (52.3%) were urban inhabitants. The frequency of donation was significantly associated with age ($p \leq 0.001$), gender ($p = 0.001$), occupation ($p \leq 0.001$), and residence ($p \leq 0.001$), but not season ($p = 0.684$) (Table 1).

### Frequency of high-risk donor behavior, stratified by sex and age of blood donors

Table 2 shows the frequency of high-risk donor behavior stratified by sex and age of the blood donors. With respect to sex, increased proportions of males than females had previously been deferred from donating blood (81/1111 [7.3%] vs 5/206 [2.4%], $p = 0.008$), were drug addicts (59/1111 [5.3%] vs 1/206 [0.5%], $p = 0.001$), had body modifications (47/1111 [4.2%] vs 2/206 [1.0%], $p = 0.025$), have had unprotected sex with multiple sexual partners (180/1111 [16.7%] vs 14/206 [6.8%], $p = 0.045$), sexually transmitted infections (185/1111 [16.7%] vs 14/206 [6.8%], $p \leq 0.001$), whereas more females than males had recently received treatment or vaccination (8/206 [3.9%] vs 17/1111 [1.5%], $p = 0.044$).

**Table 2. Frequency of high-risk donor behavior stratified by sex and age of blood donors at Mankranso Government Hospital, Ghana (2017–2020).**

| Variables | Sex | | p | Age group (in years) | | | p |
|---|---|---|---|---|---|---|---|
| | Male (*n* = 1111) | Female (*n* = 206) | | 17–26 (*n* = 575) | 27–36 (*n* = 476) | 37–60 (*n* = 266) | |
| | Yes | Yes | | Yes | Yes | Yes | |
| Are you feeling unwell today? | 71 (6.4) | 18 (8.7) | 0.226 | 44 (7.7) | 31 (6.5) | 14 (5.3) | 0.423 |
| Have you ever been deferred from donating blood? | 81 (7.3) | 5 (2.4) | **0.008** | 29 (5.0) | 34 (7.1) | 23 (8.6) | 0.115 |
| Are you taking drugs for high BP, diabetes, or asthma? | 8 (0.7) | 2 (1.0) | 0.661 | 4 (0.7) | 2 (0.4) | 4 (1.5) | 0.257 |
| Do you have history of tuberculosis, epilepsy, ulcer? | 30 (2.7) | 11 (5.3) | 0.076 | 15 (2.6) | 10 (2.1) | 16 (6.0) | **0.009** |
| Have you been treated or vaccinated in the last 4 weeks? | 17 (1.5) | 8 (3.9) | **0.044** | 14 (2.4) | 4 (0.8) | 7 (2.6) | 0.104 |
| Do you have jaundice, liver disease, or hepatitis? | 7 (0.6) | 0 (0.0) | 0.604 | 3 (0.5) | 1 (0.2) | 3 (1.1) | 0.257 |
| Do you have sickle cell disease or thalassemia? | 8 (0.7) | 0 (0.0) | 0.619 | 2 (0.3) | 3 (0.6) | 3 (1.1) | 0.399 |
| Are you a drug addict? | 59 (5.3) | 1 (0.5) | **0.001** | 25 (4.3) | 22 (4.6) | 13 (4.9) | 0.937 |
| Have you had a tattoo or cutting by traditional healer, including circumcision in the last 12 months? | 47 (4.2) | 2 (1.0) | **0.025** | 26 (4.5) | 16 (3.4) | 7 (2.6) | 0.353 |
| Have you had dental treatment in the last 1 week? | 10 (0.9) | 0 (0.0) | 0.377 | 3 (0.5) | 2 (0.4) | 5 (1.9) | 0.061 |
| Have you had a major surgery in the past 6 months? | 1 (0.1) | 0 (0.0) | 1.000 | 0 (0.0) | 1 (0.2) | 0 (0.0) | 0.413 |
| Have you received blood transfusion in the last 6 months? | 1 (0.1) | 0 (0.0) | 1.000 | 1 (0.2) | 0 (0.0) | 0 (0.0) | 0.524 |
| Have you lost >5kg due to illness in the last 6 months? | 25 (2.3) | 6 (2.9) | 0.614 | 21 (3.7) | 10 (2.1) | 0 (0.0) | **0.005** |
| Have you had unprotected sex with multiple partners or have paid/been paid to have sex in the last 6 months? | 180 (16.2) | 22 (10.7) | **0.045** | 88 (15.3) | 76 (16.0) | 38 (14.3) | 0.830 |
| Have you tested positive for gonorrhea or other STIs? | 185 (16.7) | 14 (6.8) | **≤0.001** | 90 (15.7) | 80 (16.8) | 29 (10.9) | 0.088 |
| Have you had sex with a homosexual in the last 6 months? | 3 (0.3) | 0 (0.0) | 1.000 | 1 (0.2) | 2 (0.4) | 0 (0.0) | 0.483 |
| Do you or your partner have HIV or hepatitis? | 7 (0.6) | 1 (0.5) | 1.000 | 4 (0.7) | 3 (0.6) | 1 (0.4) | 0.855 |
| After blood donation, will you do any rigorous activity? | 36 (3.2) | 2 (1.0) | 0.108 | 18 (3.1) | 11 (2.3) | 9 (3.4) | 0.632 |
| Are you pregnant? | n/a | 14 (6.8) | n/c | 8 (1.4) | 6 (1.3) | 0 (0.0) | **0.026** |

The data are presented as frequencies with respective column percentages in parentheses; Fisher's Exact test and Pearson Chi-Square were used to compare the differences in proportions between sex and age groups, respectively across different high-risk donor behaviors; BP: Blood pressure; STIs: Sexually transmitted infections; HIV: Human immunodeficiency virus; kg: Kilogram; n/a: Not applicable; n/c: Not computed; *p* was significant at ≤0.05.

Increased proportion (16/266 [6.0%]) of the prospective blood donors within the 37–60-year range had history of tuberculosis, followed by 17–26 years (15/575 [2.6%]), and 27–36 years (10/476 [2.1%]). History of tuberculosis infection was significantly associated with age group (*p* = 0.009). Significantly increased proportion of donors in the 17–26-year group than the 27–36-year group had history of >5kg body weight loss due to illness (21/575 [3.7%] vs 10/476 [2.1%], *p* = 0.005). Similarly, increased proportion of donors in the 17–26-year group than the 27–36-year group were pregnant (4/575 [1.4%] vs 6/476 [1.3%], *p* = 0.026) (Table 2).

## Frequency of high-risk donor behavior among the prospective blood donors stratified by frequency of donation and season

Table 3 shows the frequency of high-risk blood donor behavior stratified by frequency of donation and season. A significantly increased proportion of first-time than repeat donors were unwell (70/856 [8.2%] vs 19/461 [4.1%], p = 0.005), had previously been deferred from donating blood (72/856 [8.4%] vs 14/461 [3.0%], p≤0.001), were drug addicts (47/856 [5.5%] vs 13/461 [2.8%], p = 0.027), had body modifications (42/856 [4.9%] vs 7/461 [1.5%], p = 0.001), have had unprotected sex with multiple sexual partners (145/856 [16.9%] vs 57/461 [12.4%], p = 0.030), had history of sexually transmitted infections (146/856 [17.1%] vs 53/461 [11.5%], p = 0.008), were going to perform rigorous activities after donation (31/856 [3.6%] vs

**Table 3. Frequency of high-risk donor behavior stratified by frequency of donation and season, at Mankranso Government Hospital (2017–2020).**

| Variables | Frequency of donation | | p | Season | | p |
|---|---|---|---|---|---|---|
| | First-time (n = 856) | Repeat (n = 461) | | Rainy (n = 741) | Dry (n = 576) | |
| | Yes | Yes | | Yes | Yes | |
| Are you feeling unwell today? | 70 (8.2) | 19 (4.1) | **0.005** | 53 (7.2) | 36 (6.3) | 0.580 |
| Have you ever been deferred from donating blood? | 72 (8.4) | 14 (3.0) | **≤0.001** | 49 (6.6) | 37 (6.4) | 0.911 |
| Are you taking drugs for high BP, diabetes, or asthma? | 4 (0.5) | 6 (1.3) | 0.107 | 7 (0.9) | 3 (0.5) | 0.527 |
| Do you have history of tuberculosis? | 27 (3.2) | 14 (3.0) | 1.000 | 21 (2.8) | 20 (3.5) | 0.526 |
| Have you been treated or vaccinated in the last 4 weeks? | 15 (1.8) | 10 (2.2) | 0.673 | 12 (1.6) | 13 (2.3) | 0.422 |
| Do you have jaundice or liver disease? | 5 (0.6) | 2 (0.4) | 1.000 | 2 (0.3) | 5 (0.9) | 0.251 |
| Do you have sickle cell disease or thalassemia? | 6 (0.7) | 2 (0.4) | 0.721 | 6 (0.8) | 2 (0.3) | 0.478 |
| Are you a drug addict? | 47 (5.5) | 13 (2.8) | **0.027** | 34 (4.6) | 26 (4.5) | 1.000 |
| Have you had a tattoo or cutting by traditional healer, including circumcision in the last 12 months? | 42 (4.9) | 7 (1.5) | **0.001** | 31 (4.2) | 18 (3.1) | 0.379 |
| Have you had dental treatment in the last 1 week? | 6 (0.7) | 4 (0.9) | 0.747 | 7 (0.9) | 3 (0.5) | 0.527 |
| Have you had a major surgery in the past 6 months? | 1 (0.1) | 0 (0.0) | 1.000 | 1 (0.1) | 0.0 | 1.000 |
| Have you received blood transfusion in the last 6 months? | 1 (0.1) | 0 (0.0) | 1.000 | 0 (0.0) | 1 (0.1) | 0.437 |
| Have you lost >5kg due to illness in the last 6 months? | 25 (2.9) | 6 (1.3) | 0.085 | 17 (2.3) | 14 (2.4) | 0.857 |
| Have you had unprotected sex with multiple partners or have paid/been paid to have sex in the last 6 months? | 145 (16.9) | 57 (12.4) | **0.030** | 103 (13.9) | 99 (17.2) | 0.106 |
| Have you tested positive for gonorrhea or other STIs? | 146 (17.1) | 53 (11.5) | **0.008** | 95 (12.8) | 104 (18.1) | **0.010** |
| Have you had sex with a homosexual in the last 6 months? | 3 (0.4) | 0 (0.0) | 0.556 | 3 (0.4) | 0 (0.0) | 0.261 |
| Do you or your partner have HIV or hepatitis? | 7 (0.8) | 1 (0.2) | 0.274 | 3 (0.4) | 5 (0.9) | 0.308 |
| After blood donation, will you do any rigorous activity? | 31 (3.6) | 7 (1.5) | **0.037** | 19 (2.6) | 19 (3.3) | 0.507 |
| Are you pregnant? | 12 (1.4) | 2 (0.2) | **0.002** | 8 (1.1) | 6 (1.0) | 1.000 |

The data are presented as frequencies with respective column percentages in parentheses; Fisher's Exact test was used to compare the differences in proportions of donation frequency and season across different high-risk donor behaviors; BP: Blood pressure; STIs: Sexually transmitted infections; HIV: Human immunodeficiency virus; kg: Kilogram; p was significant at ≤0.05.

7/461 [1.5%], p = 0.037), and were pregnant (12/856 [1.4%] vs 2/461 [0.2%], p = 0.002). Furthermore, a significantly increased proportion of the prospective donors screened in the dry season than rainy season had previously tested positive for gonorrhea and other sexually transmitted infections (104/576 [18.1%] vs 95/741 [12.8%], p = 0.010), whereas no association existed between season and the other risk behaviors (Table 3).

## Levels of hemoglobin and anthropometric characteristics of the blood donors, stratified by frequency of donation

Fig 1 shows the median hemoglobin, weight, systolic- and diastolic BP of the prospective donors. The level of hemoglobin (13.5 vs 13.8, p = 0.0032), weight (64.0 vs 65.8, p = 0.0003), and diastolic BP (79.0 vs 80.0, p = 0.0241) were significantly reduced in the first-time donors than in the repeat donors. However, the median systolic BP did not differ between first-time and repeat donors (120.0 vs 120.0, p = 0.9209) (Fig 1).

## Association between hemoglobin level and sexually transmitted infections

Fig 2 shows the association between the level of hemoglobin and history of STIs. Out of 199 donors that responded 'yes' to history of gonorrhea and other STIs, 17.6% (36) had low

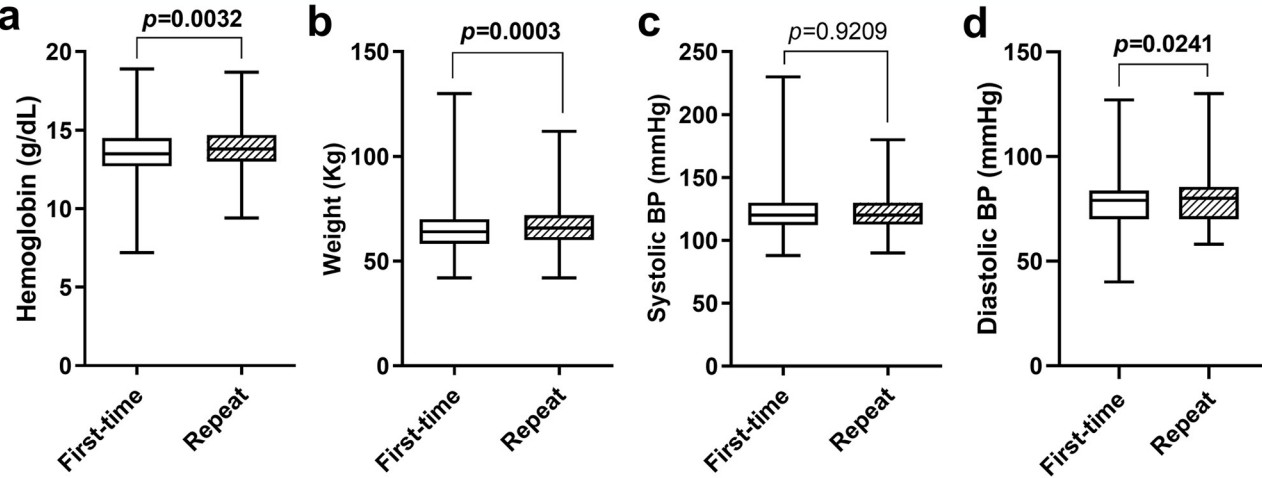

**Fig 1. Levels of hemoglobin and anthropometric characteristics of blood donors, stratified by frequency of blood donation.** The data are presented as boxplots and whiskers showing 25th, 50th, and 75th quartiles; The Mann-Whitney U test was used to compare differences in medians of hemoglobin and anthropometric characteristics between first-time and repeat blood donors; BP: Blood pressure; *p* was significant at ≤0.05.

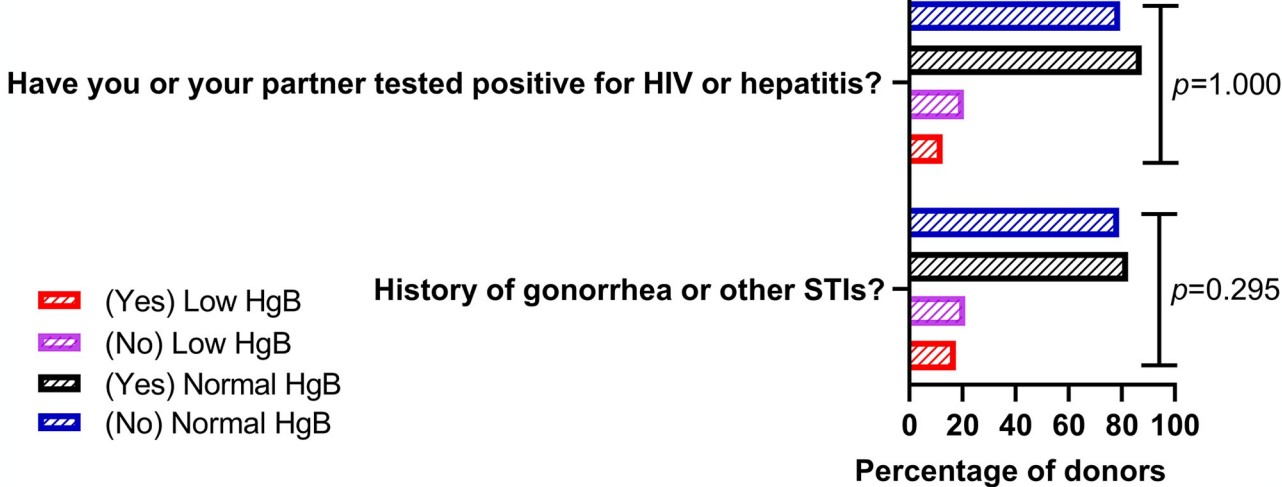

**Fig 2. Association between level of hemoglobin and sexually transmitted infections.** HgB: Hemoglobin; STI: Sexually transmitted infection; $p \leq 0.05$ was considered significant for all analyses.

hemoglobin, while 82.4% (164) had normal hemoglobin level. Conversely, out of the 1118 donors that responded 'no', 21.0% (235) had low hemoglobin, while 79.0% (883) had normal hemoglobin. Furthermore, out of 8 donors who responded 'yes' to the question 'have you or your partner tested positive for HIV or hepatitis?', 12.5% (1) had low hemoglobin, while 87.5% (7) had normal hemoglobin. However, out of the 1309 donors that responded 'no', 20.6% (269) had low hemoglobin, while 79.4% (1040) had normal hemoglobin level. Hemoglobin level had no significant association with both history of gonorrhea/other STIs, and either the donors or their partners having a history of positive HIV or hepatitis (*p* = 0.295 vs *p* = 1.000, respectively) (Fig 2).

## Discussion

High-risk behavior and practices perpetrated by blood donors could compromise blood safety owing to the existence of the immunological incubation period. The current study, therefore, determined the frequency and patterns of self-reported high-risk behavior among replacement blood donors that could negatively impact the safety of blood in Ghana. In the current study, most (43.7%) of the prospective donors were significantly young and in their prime (17–26 years). The frequency of the donors then decreased consistently with increasing age. These were similar to the findings of studies from Ghana [11] and Nepal [12]. A similar pattern was observed in the first-time donors when stratified by donation frequency, which is consistent with the findings of Asamoah-Akuoko et al., [11]. In Ghana, many young individuals with ages 17–26 years are likely in high school, college/university or undergoing national service after tertiary education, and are therefore, still dependent on their parents/guardians. Also, in many Ghanaian societies, young individuals are forbidden from questioning the authority of elders. Therefore, it is plausible that such individuals unwillingly participated in replacement blood donations out of fear of losing financial support from parents/guardians and to prevent being stigmatized by other relatives. Conversely, the majority (40.8%) of the repeat donors were significantly older, with the majority between 27 and 36 years. These donors are relatively matured compared to the 17–26-year group, and therefore have a better understanding of the value of blood, and what their social obligations are in regard to ensuring sustainability of the blood supply chain. Conversely, the low participation of much older individuals (37–60 years), is probably because, they are prone to risk factors like obesity that predisposes them to other chronic non-communicable diseases like diabetes, hypertension, and deranged cardiac function, all of which are contraindications for blood donation.

The donors were predominantly (84%) males, and culminated in a 5:1 male-to-female ratio, with similar patterns observed in both first-time and repeat donors. The observation of more male than female ($p$ = 0.001) donors in this study was consistent with findings of several studies conducted in Ghana [1, 6, 9, 11, 12], and other parts of Africa [13–16]. In many societies, men are socially obliged and responsible for the safety and wellbeing of their family members [2], which could have resulted in the increased participation of males in blood donation. Conversely, the low number of female donors is attributed to pregnancy and childbearing-related characteristics such as menstruation, and breastfeeding, which render females less suitable donor candidates compared to males [1]. Furthermore, some studies [10, 15] suggest that the frequent loss of blood through menstruation could culminate in anemia, and this rather increases males' prospects for blood donation than females.

The observation of more informal sector workers ($p \leq 0.001$) among the blood donors in this study corroborates the findings of a study from Gabon [17]. It is worthy to note that, the current study area was a rural setting where majority of the inhabitants were engaged in non-formal occupations like farming, trading, artisanal jobs, and illegal mining (known locally as galamsey). This may have accounted for the significantly increased frequency of both informal sector workers ($p \leq 0.001$), and rural inhabitants ($p \leq 0.001$) in the donor population. More donors were recorded in the rainy- than the dry season, probably due to the increased demand for blood transfusion in the rainy season [5], and although not different between first-time and repeat donors, the relationship between seasonal variation and certain transfusion-associated diseases have been established [18].

The observation of more male than female drug addicts in this study was consistent with existing literature, which suggests that although substance abuse is considered a deviant behavior and therefore associated with public disapproval, more females than males are stigmatized for their involvement [19]. Furthermore, men are believed to have more social exposure to

such social vices than females [19]. In recent years, aesthetic body enhancements like tattoos, and piercings have gained popularity in Ghana. In this study, more males than females had undergone permanent body modifications, which is consistent with the findings of Heywood *et al*., [20]. Males are more likely to belong to gang groups, in which a display of group tattoos proves membership [21] and loyalty. Also, according to Weiler *et al*., [22], tattoos in particular are used by many to express one's unique identity. Furthermore, certain body modifications like tribal marks, and cuttings by traditional healers are particularly confined to some ethnic groups in Ghana and Africa. Therefore, the descent of the population could have contributed to the frequency of body modifications among the respondents.

The observation of more males with multiple sexual partners corroborates the findings of Dendup *et al*., [23] from Bhutan, although, their study involved a slightly younger population. Maintaining female purity ("virginity until marriage") [23] and monogamy are among some social norms expected of females in many Ghanaian settings, and could have contributed to the reduced involvement of females with multiple sexual partners. Dendup *et al*., [23] report that males having sex with multiple sexual partners is a way of expressing male masculinity and dominance. Furthermore, the increased illegal mining activities in the study district may have contributed to this, as it has attracted many people from different parts of the country, most of whom are males and individuals with low- or no level of formal education.

In the present study, self-reported history of sexually transmitted infections (STIs) was significantly associated with the sex of respondents, with more males than females reporting STIs. This finding is consistent with studies from Ghana [1, 11] and Ethiopia [16] that determined the burden of STIs among blood donors in which more males were infected with blood-borne STIs. Scientific evidence suggests a positive correlation between immunity and the female sex, with females less infected by viral pathogens [1]. The high concentration of estrogen in females enable them to produce sufficient resistance to viral pathogens by synthesizing more CD4+ cells that trigger a marked T cell response against viruses [1, 24]. Furthermore, this observation could be attributed to the increased involvement of males in other predisposing high risk behaviors like sex with multiple partners, use of self-injected drugs, and body modifications [20], all of which were common among males in this study.

A self-reported history of tuberculosis (TB) was significantly associated with age, with older respondents (37–60 years) reporting previous TB infection. This finding corroborates a study from Ghana [25]. The possible reason is that immunity in humans diminishes with ageing, which may result from a reduced capacity to produce cytokines and restore old T lymphocytes [25]. In Ghana, the ages, 17–26 years are typical of individuals in high school or other tertiary institutions. Therefore, the frequent weight loss in this population could be ascribed to unhealthy lifestyles, and lack of monitoring due to the absence of parents in schools. Also, the weight loss may have been motivated by looks and peer pressure [26]. Furthermore, it could be due to malnutrition, since the predominant male population is usually dependent on females for their nutritional needs. Pregnancy was frequent among teenagers and younger adults, since this population is considered a vulnerable group.

It is plausible that first-time replacement blood donors have limited to no knowledge regarding the requirements for blood donation compared to repeat donors, who may have completed the donor selection questionnaire in the past, and therefore, are familiar with the recruitment process. This may have accounted for the significantly increased frequency of some self-reported risk behaviors among the first-time donors. For instance, increased proportion of them were unwell ($p = 0.005$), had been deferred from donating in the past ($p \leq 0.001$), were drug addicts ($p = 0.027$), had body modifications ($p = 0.001$), have had unprotected sex with multiple sexual partners ($p = 0.030$), had tested positive for STIs ($p = 0.008$), had impending rigorous activities following blood donation ($p = 0.037$), and were pregnant ($p = 0.002$).

This could explain the increased hemoglobin and body weight among the repeat donors compared to first-time donors. Therefore, it is imperative that repeat donations are encouraged even during first-time donations, since such donors are familiar with the donor selection criteria, and are likely to self-defer if they fail to meet the requirements. However, the observation of a significantly increased self-reported STIs among replacement donors in the dry season ($p = 0.010$) is not well understood.

The study was not without limitations: for instance, the practice of tribal marks is common particularly among some ethnic groups in northern Ghana, therefore, information about ethnicity would have explained the increased reports of body modifications. Also, the study did not present the nexus between high-risk behavior and donor adverse events. Furthermore, the retrospective nature of the study did not permit decoupling of the various STIs. We, therefore, recommend that future studies should decouple the STIs and consider collecting data on blood donor adverse events to fill this knowledge gap.

## Conclusion

The frequency of blood donation was significantly associated with age, sex, occupation, and residence of the donors, with first-time donors younger than repeat donors. Deferral from blood donation, drug addiction, body modification, multiple sexual partners, and positive STIs were frequent among male- than female donors, whereas, vaccination was frequent among females than males. A history of tuberculosis was frequently self-reported by older adults, whereas, weight loss and pregnancy were frequent among teenagers and younger individuals. An increased proportion of first-time replacement- than repeat donors reported being unwell, deferred from donating, drug addicts, had body modifications, had multiple sexual partners, were positive for STIs, and were pregnant during the screening. Hemoglobin, weight, and diastolic BP were significantly reduced among first-time donors.

## Supporting information

**S1 Data. Dataset used for the study.**
(SAV)

## Acknowledgments

The authors wish to thank the Medical Superintendent, and all Laboratory staff of Mankranso Government Hospital for their support.

## Author Contributions

**Conceptualization:** Felix Osei-Boakye.

**Data curation:** Felix Osei-Boakye, Charles Nkansah, Samuel Kwasi Appiah, Gabriel Abbam, Charles Angnataa Derigubah.

**Formal analysis:** Felix Osei-Boakye.

**Methodology:** Felix Osei-Boakye, Charles Nkansah, Samuel Kwasi Appiah, Gabriel Abbam, Charles Angnataa Derigubah.

**Project administration:** Felix Osei-Boakye.

**Resources:** Felix Osei-Boakye.

**Supervision:** Felix Osei-Boakye.

**Validation:** Felix Osei-Boakye, Charles Nkansah, Ejike Felix Chukwurah.

**Visualization:** Felix Osei-Boakye.

**Writing – original draft:** Charles Nkansah, Samuel Kwasi Appiah, Gabriel Abbam, Boniface Nwofoke Ukwah, Victor Udoh Usanga, Emmanuel Ike Ugwuja, Ejike Felix Chukwurah.

**Writing – review & editing:** Charles Nkansah, Samuel Kwasi Appiah, Gabriel Abbam, Boniface Nwofoke Ukwah, Victor Udoh Usanga, Emmanuel Ike Ugwuja, Ejike Felix Chukwurah.

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
