## [Decision Letter · Decision Letter 0]

11 Jun 2024

PONE-D-24-15733Self-reported high-risk behavior among first-time and repeat replacement blood donors; a four-year retrospective study of patternsPLOS ONE

Dear Dr. Osei-Boakye,

Thank you for submitting your manuscript to PLOS ONE. After careful consideration, we feel that it has merit but does not fully meet PLOS ONE’s publication criteria as it currently stands. Therefore, we invite you to submit a revised version of the manuscript that addresses the points raised during the review process.

We look forward to receiving your revised manuscript.

Kind regards,

Theresa Ukamaka Nwagha, M.B.B.S., M.P.H., FMCPath, M.Sc., M.D.,

Academic Editor

PLOS ONE

Reviewers' comments:

Reviewer's Responses to Questions

**Comments to the Author**

1. Is the manuscript technically sound, and do the data support the conclusions?

Reviewer #1: Yes

Reviewer #2: Yes

2. Has the statistical analysis been performed appropriately and rigorously? 

Reviewer #1: Yes

Reviewer #2: Yes

3. Have the authors made all data underlying the findings in their manuscript fully available?

Reviewer #1: Yes

Reviewer #2: Yes

4. Is the manuscript presented in an intelligible fashion and written in standard English?

Reviewer #1: No

Reviewer #2: Yes

5. Review Comments to the Author

Reviewer #1: The manuscript has the information but the way it was presented could have been better. The title 'Study design and setting' has the year the study was commenced omitted making it look like the study was conducted only in 2024. This column should be re-written for fluency. Check and correct dates throughout the text.

Under the heading study population. Family replacement donors undergo screening for blood donation not transfusion as stated since they are the donors not patients.

The heading pre- screening of blood donors and administration of questionnaire made the study look like a prospective rather than a retrospective study. This column can be removed, and the information summarized into the data collection column and stated retrospectively.

Reviewer #2: Why only Tuberculosis is add in the high risk behaviour other than STD's? Why not other infections?

Why no association of Haemoglobin has been seen any where? I think you should also give the haemoglobin of those who had STD's

STD's are given generally, why not specific STD is mentioned during history taking?

Were female donors asked properly about STD's history? Were female phlebotomist/ doctors available for their history?

Which guidelines are followed at GHANA for UDHQ? As per AABB, if tattoo is now done by a closed system, donor is accepted

Is there any criteria of reentry for such donors specially repeat donors?

6. PLOS authors have the option to publish the peer review history of their article (what does this mean?). If published, this will include your full peer review and any attached files.

Reviewer #1: No

Reviewer #2: **Yes: **Samra Waheed

---

## [Author Response · Author response to Decision Letter 0]

22 Jul 2024

The manuscript has been formatted according to the PLOS ONE style.

All relevant data are within the manuscript and its Supporting Information files.

We recommend that you contact the original copyright holder with the Content Permission Form (http://journals.plos.org/plosone/s/file?id=7c09/content-permission-form.pdf) and the following text: “I request permission for the open-access journal PLOS ONE to publish XXX under the Creative Commons Attribution License (CCAL) CC BY 4.0 (http://creativecommons.org/licenses/by/4.0/). Please be aware that this license allows unrestricted use and distribution, even commercially, by third parties. Please reply and provide explicit written permission to publish XXX under a CC BY license and complete the attached form.” Please upload the completed Content Permission Form or other proof of granted permissions as an "Other" file with your submission. In the figure caption of the copyrighted figure, please include the following text: “Reprinted from [ref] under a CC BY license, with permission from [name of publisher], original copyright [original copyright year].”

Maps at the CIA (public domain): https://www.cia.gov/library/publications/the-world-factbook/index.html and

https://www.cia.gov/library/publications/cia-maps-publications/index.html

Figure 1 has been removed from the manuscript. 

The references have been updated.

Reviewer #1: 

1. The manuscript has the information but the way it was presented could have been better. The title 'Study design and setting' has the year the study was commenced omitted making it look like the study was conducted only in 2024. This column should be re-written for fluency.

In this study, we retrospectively reviewed and collected archived data spanning 2017 and 2020. And, this was done as a onetime sampling in 2024 (from 14th March to 4th April 2024). So, for the purpose of clarity, the study was conducted only in 2024, using archived/retrospective data. This information has been modified and the clarity has been improved. This can be found on page 6, line 101-104.

2. Check and correct dates throughout the text.

All the dates have been reformatted.

3. Under the heading study population. Family replacement donors undergo screening for blood donation not transfusion as stated since they are the donors not patients.

The sentence has been corrected.

4. The heading pre- screening of blood donors and administration of questionnaire made the study look like a prospective rather than a retrospective study. This column can be removed, and the information summarized into the data collection column and stated retrospectively.

The section has been removed and the necessary changes have been effected.

Reviewer #2: 

1. Why only Tuberculosis is add in the high risk behaviour other than STD's? Why not other infections?

History of sexually transmitted infections have been reported in the lower part of both tables 2 and 3 on pages 12 and 14, respectively and figure 2 on page 16.

- Have you tested positive for gonorrhea or other STIs?

- Do you or your partner have HIV or hepatitis? 

2. Why no association of Haemoglobin has been seen any where? I think you should also give the haemoglobin of those who had STD's

There was no significant association between haemoglobin and sexually transmitted infections. This has been included in the revised manuscript as Figure 2, on page 16, line 302-318.

3. STD's are given generally, why not specific STD is mentioned during history taking?

The retrospective nature of the study did not permit modification of the questions and or decoupling of the STIs. This has been captured as a limitation to the study, on page 21, line 426-428.

4. Were female donors asked properly about STD's history? Were female phlebotomist/ doctors available for their history?

This was a retrospective study, and only data from archived records were accessible. Consequently, the researchers did not play any role in the administration of the questionnaires.

5. Which guidelines are followed at GHANA for UDHQ? As per AABB, if tattoo is now done by a closed system, donor is accepted

Is there any criteria of reentry for such donors specially repeat donors?

The document used in Ghanaian blood centers that is similar to the UHDQ is the ‘Donor Clinical Record’. This is available at https://figshare.com/s/69f8efdc1ddaf1ec6dd5.

Donors with tattoos are permitted to donate blood after six months, if they test negative for STIs, and also, satisfy other pre-screening conditions. This is because hepatitis B virus which is among the infections screened for has an incubation period of about six months.

---

## [Editor Report · Decision Letter 1]

24 Jul 2024

Self-reported high-risk behavior among first-time and repeat replacement blood donors; a four-year retrospective study of patterns

PONE-D-24-15733R1

Dear Dr. Osei-Boakye,

We’re pleased to inform you that your manuscript has been judged scientifically suitable for publication and will be formally accepted for publication once it meets all outstanding technical requirements.

Kind regards,

Theresa Ukamaka Nwagha, M.B.B.S., M.P.H., FMCPath, M.Sc., M.D.,

Academic Editor

PLOS ONE
---

## [Editor Report · Acceptance letter]

31 Jul 2024

PONE-D-24-15733R1 

PLOS ONE

Dear Dr. Osei-Boakye, 

I'm pleased to inform you that your manuscript has been deemed suitable for publication in PLOS ONE. Congratulations! Your manuscript is now being handed over to our production team.

Kind regards, 

on behalf of

Dr. Theresa Ukamaka Nwagha 

Academic Editor

PLOS ONE